# A case series shows independent vestibular labyrinthine function after major surgical trauma to the human cochlea

Stefan K. Plontke [1✉], Torsten Rahne[1], Ian S. Curthoys [2], Bo Håkansson [3] & Laura Fröhlich[1]

## Abstract

**Background** The receptors for hearing and balance are housed together in the labyrinth of the inner ear and share the same fluids. Surgical damage to either receptor system was widely believed to cause certain permanent loss of the receptor function of the other. That principle, however, has been called into question because there have been anecdotal reports in individual patients of at least partial preservation of cochlear function after major surgical damage to the vestibular division and vice versa.

**Methods** We performed specific objective vestibular function tests before and after surgical trauma (partial or subtotal cochlear removal) for treatment of intracochlear tumors in 27 consecutive patients in a tertiary referral center. Vestibular function was assessed by calorics (low-frequency response of the lateral semicircular canal), vestibulo-ocular reflex by video head impulse test (vHIT) of the three semicircular canals, cervical and ocular vestibular evoked myogenic potentials (cVEMP, saccule and oVEMP, utricle). Preoperative and post-operative distributions were compared with paired $t$-tests.

**Results** Here we show that there was no significant difference between pre- and post-operative measures for all tests of the five vestibular organs, and that after major surgical cochlear trauma, the vestibular receptors continue to function independently.

**Conclusions** These surprising observations have important implications for our under-standing of the function and the surgery of the peripheral auditory and vestibular system in general and open up new possibilities for the development, construction and evaluation of neural interfaces for electrical or optical stimulation of the peripheral auditory and vestibular nervous system.

## Plain language summary

Receptors for the hearing and balance systems are located in the inner ear. These are very delicate structures that convert sound and movement into nerve signals to control hearing and balance. Surgical damage to either system was widely believed to cause loss of function in the other. Here, we investigated the function of the balance receptors before and after severe damage to the part of the inner ear which is responsible for hearing (the cochlea) due to surgical removal of tumurs. We show that despite the close proximity of both types of receptors and the severe trauma to the cochlea, in most patients the balance system can still function normally. This observation may have important implications for how we treat patients with inner ear surgery and for the techniques we use to treat hearing and balance disorders.

[1] Department of Otorhinolaryngology, Head & Neck Surgery, Martin Luther University Halle-Wittenberg, University Medicine Halle, Halle (Saale), Germany. [2] Vestibular Research Laboratory, School of Psychology, The University of Sydney, Sydney, NSW, Australia. [3] Chalmers University of Technology, Electrical Engineering, Gothenburg, Sweden. ✉email: stefan.plontke@uk-halle.de

The perception of balance and spatial orientation is achieved and maintained by a complex set of sensorimotor control systems including input from the vestibular receptors in the inner ear. Diseases or trauma with disturbances of vestibular function result in dizziness and vertigo and exhibit a substantial impact on daily life in humans, especially for bilateral vestibular hypofunction[1]. The receptors for both the sense of hearing and balance—the cochlear and vestibular sensory systems—are housed together in the membranous labyrinth of the inner ear and share the same fluids with a tightly regulated homeostasis. The fluid bathing the receptors of both systems, endolymph, is crucial for normal receptor function and the major generation of this fluid takes place in the stria vascularis of the cochlea.

Surgical approaches to each of these systems have adhered to a principle, that damage to the membranous labyrinth of either system will cause almost certain permanent loss of the receptor function of both systems. Although the overall clinical effect of minimally invasive surgical approaches to the cochlea (like a cochleostomy during cochlear implantation, CI) on the vestibular function is considered to be non-significant[2], CI may lead to vestibular dysfunction and dizziness in some patients[2–5]. Therefore, any inner ear surgery generally aims on 'atraumatic' surgical techniques. That principle, however, has been called into question because there have been anecdotal reports in individual patients of at least partial preservation of cochlear function after major surgical damage to the vestibular division and vice versa[6–13].

Here we report for the first time (to the best of our knowledge) in a large case series, that after major trauma to the cochlea—cochlear removal to treat intracochlear schwannoma, removing the membranous labyrinth of the cochlea—the vestibular receptors continue to function normally, as shown by specific, objective tests of peripheral vestibular function before and after surgery.

These results demonstrate that under particular conditions, the cochlear and vestibular sensory systems can function independently, which has important implications for inner ear surgical procedures, and the development of neural interfaces of sensory prostheses for the auditory system and gene and cell-based inner ear therapies.

## Methods

**Study design, setting, and participants**. The study comprises a large retrospective and prospective monocentric case series analyzing results from a tertiary (university) referral center. Patients included all consecutive patients with surgical resection of intracochlear schwannomas, most of them with hearing rehabilitation with a cochlear implant (CI).

**Surgery**. Surgical tumor removal was achieved through partial or subtotal cochleoectomy as described in detail elsewhere[14]. In short: The tumor was removed through a retroauricular-transmeatal, microscopic surgical approach. The cochlear capsule was opened exposing the tumor anterior to the RW, followed by drilling along the basal turn. A bony arch was left at the round window to later secure the electrode carrier (Fig. 1). The second turn was opened anterior to the oval window. In order to gain access to tumor parts located medially to the modiolus, the apical parts of the modiolus were removed preserving as much as possible of the modiolus, which contains the spiral ganglion cells in Rosenthal's canal. During surgery, the base of scala tympani was blocked towards the hook region and the base of scala vestibuli and scala media with soft tissue. If patients received a CI, a device with a precurved, perimodiolar electrode carrier (CI512 or CI612; Nucleus CI, Cochlear, Sydney, Australia) was used. The cochlear defect was closed with an autologous cartilage/perichondrium compound transplant and bone pâté.

**Measurements of peripheral vestibular and auditory function**. Prior to the surgery (<6 months) and during 12 months after surgery, the function of each vestibular sensory region was objectively assessed by specific, objective peripheral vestibular tests: Calorics (low-frequency response of the lateral semicircular canal), vestibulo-ocular reflex (VOR) by video head impulse test (vHIT) (manually induced head impulse response of lateral, anterior and posterior semicircular canals), cervical vestibular-evoked myogenic potentials (cVEMP, saccule), and ocular vestibular-evoked myogenic potentials (oVEMP, utricle). The function of the auditory system was assessed as speech recognition with the CI, if applicable.

*Spontaneous nystagmus and calorics*. Videonystagmography was performed using a Hortmann Vestlab videonystagmography system and the Vestlab OS software (GN Otometrics, Taastrup, Denmark). Over a recording period of 30 s, eye movements larger than 0.3°/s were identified and considered as spontaneous nystagmus (SPN) and counted. Caloric stimulation of both external ear canals was maintained by cold (30 °C) and warm (44 °C) water volumes of 75 ml within time periods of 30 s. Low-frequency responses of the lateral semicircular canal (SCC) with horizontal and vertical eye movements were recorded for 60 s. Eye movements larger than 0.75°/s were identified and considered as nystagmus. Slow-phase velocity (SPV) of all eye movements after ipsilateral and contralateral caloric irrigation was automatically calculated. Maximum absolute SPV values of the ipsilateral ($SPV_{ipsi}$) and contralateral ($SPV_{contra}$) sides were used to calculate the vestibular response relative to the side affected by the tumor with the formula: $(SPV_{ipsi,cold} + SPV_{ipsi,warm} - SPV_{contra,cold} - SPV_{contra,warm})/(SPV_{ipsi,cold} + SPV_{ipsi,warm} + SPV_{contra,cold} + SPV_{contra,warm})$. Negative values indicated a reduced response on the ipsilateral side, and positive values indicated a reduced response on the contralateral side. Results below –25% and above 25% were considered abnormal.

*Vestibulo-ocular reflex*. VOR recordings were performed to assess impulse responses of all SCCs using a video head-impulse test (vHIT) system (GN Otometrics, Taastrup, Denmark). After small (10–20°) and fast unpredictable passive head turns aligned to the SCC orientations (lateral, left anterior-right posterior (LARP), and right anterior-left posterior (RALP) planes), the vestibulo-ocular response (VOR gain) was measured together with overt and covert saccades which were identified after recording by a 250-Hz high-speed eye camera that also measured head movements. Absolute mean VOR gain between eye and head movement of the affected and non-affected sides was measured.

*Vestibular-evoked myogenic potentials*. Otolith function was assessed by measuring cervical (cVEMP) and ocular (oVEMP) vestibular-evoked myogenic potentials using the Eclipse Platform (Interacoustics, Copenhagen, Denmark) that stimulated monaurally with tone bursts (1 or 2 cycles plateau; no rise/fall; 8 Hz stimulus repetition rate). For air-conducted VEMPs, insert earphones were used with tone bursts of 500 Hz at a stimulation level of 100 dB nHL. For bone-conducted VEMPs either a B81 transducer (Radioear, New Eagle, USA) stimulating with 500 Hz at 70 dB nHL or the recently developed B250 transducer (Chalmers, Gothenburg, Sweden) stimulating with 250 Hz at 80 dB nHL were used[15,16]. Preoperatively, only air-conducted cVEMPs and oVEMPs were measured because bone conduction was not available in most cases. Postoperative cVEMPs and oVEMPs were always measured to bone-conducted stimulation, because the incus was removed during surgical tumor removal (Fig. 1b), resulting in loss of air conduction. In addition, air-conducted VEMP testing in CI patients bears the risk of false-negative results

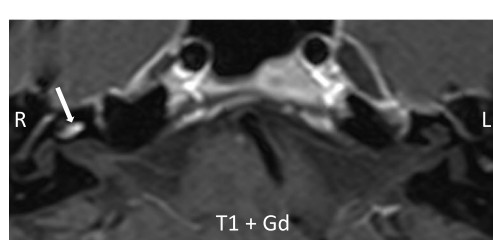
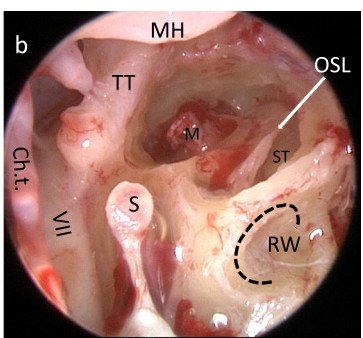

**Fig. 1 Intracochlear tumor and situs after subtotal cochlear removal. a** Axial MRI (T1-w with Gd- enhancement) showing an intracochlear schwannoma (→) in the right cochlea. Although being a very rare disease, we can gain interesting insights into inner ear (patho)physiology from its diagnostics and treatment. MRI: Dr. Georg Eisele, Radiologisches Zentrum, Wangen, Germany (with permission). **b** Intraoperative endoscopic view after tumor resection through subtotal cochleoectomy and before cochlear implantation and defect reconstruction. During surgery, the bony arch (dashed line) of the round window is preserved for securing the electrode[14]. The Ductus reuniens courses along that bony arch of the round window and in the depth of the hook region connects the cochlear duct with the saccule in the vestibule (Fig. 2),[33]. Ch. t. chorda tympani, Gd Gadolinium, L left, M modiolus (2nd cochlear turn), MH malleus handle, OSL osseus spiral lamina, R right, RW round window, S stapes, ST scala tympani (basal cochlear turn), TT tensor tympani muscle, VII facial nerve, w- weighted.

due to (inner ear) conductive loss, and response rates to bone-conducted stimulation have been reported to be higher compared to air-conducted stimulation[17]. Due to these methodological differences, only the postoperative VEMPs were analyzed, and amplitudes and latencies of the affected ipsilateral side were compared to the contralateral side as a reference.

cVEMPs were measured using self-adhesive Neuroline 720 surface electrodes (Ambu A/S, Ballerup, Denmark) placed on the upper half of the ipsilateral sternocleidomastoid muscles, a reference electrode on the sternum, and the ground electrode on the forehead. In an upright sitting position, patients were instructed to rotate their heads toward the non-stimulated ear. The electromyogram (EMG) was measured in a −20 to 80 ms window relative to the onset of the stimulus and bandpass filtered between 10 and 1000 Hz. Visual feedback was given to the patient to maintain constant muscle tension. The first positive–negative peak (p13–n23) of the averaged EMG was defined as the cVEMP amplitude and normalized by the mean root-mean-square EMG level. For every presentation, at least 200 stimuli were averaged. For oVEMP recordings, electrode pairs were placed as bipolar montage on the infra-orbital ridge 1 cm below the lower eyelid contralateral to the stimulated side and about 2 cm caudal to the first electrode with the ground electrode placed on the forehead. The patients were asked to look maximally upwards. The first negative-positive peak (n10–p15) of the averaged EMG was defined as the oVEMP amplitude. For every presentation, at least 100 stimuli were averaged. For cVEMPs and oVEMPs, amplitudes and latencies between the tumor-affected side and the non-affected side were analyzed.

*Word recognition with CI.* For patients receiving a CI, hearing performance was measured at first fit, and around 1, 3, 6, and 12 months after surgery. Word recognition scores were measured at a sound pressure level of 65 dB SPL (WRS$_{65}$) by the Freiburg monosyllables and the Freiburg numbers test after blocking and masking the contralateral ear with broad band noise, if applicable.

**Statistics and reproducibility**. If data could not be obtained in the defined postoperative test period of 12 months, they were completed by later recordings or treated as missing data. The respective *n* was reported. Preoperative and postoperative distributions were compared with paired *t*-tests. Bonferroni correction was applied if multiple comparisons were performed. The significance level was set to 5%. For all statistical calculations,

GraphPad Prism software (Version 8, Graphpad Software, San Diego, CA, USA) was used.

**Ethical approval**. The study was approved by the responsible ethics committee (approval numbers 2019-026 and 2019-050). All relevant ethical regulations were followed, informed consent was obtained from all participants.

**Reporting summary**. Further information on research design is available in the Nature Research Reporting Summary linked to this article.

## Results

**Participants**. Between 2011 and 2020, 27 patients (29–75 years, 13 male, 14 female) with solely intracochlear schwannomas underwent surgical tumor removal (Fig. 1), 23 with single stage and 2 with second stage cochlear implantation. Preoperatively, 25 patients suffered from complete or functional deafness (indication for CI). Two patients would have benefited from a hearing aid but could not use it due to increase of tinnitus and dysacusia. Hearing loss was due to sudden hearing loss in 6 of 27 (22%) or progressive or recurrent sudden hearing loss in 21 of 27 (78%) patients. Four of 27 patients (15%) reported vertigo attacks in their medical history, with two patients having been hospitalized due to their attacks. Five of 27 patients (19%) reported episodes of instability. Eighteen patients complained about tinnitus (four patients with pulsating tinnitus), while information about tinnitus was not available for nine patients.

Postoperative dizziness and vertigo varied between none to lasting for ~2 weeks. Only one patient continuously suffers from incompletely compensated ipsilateral loss of vestibular function 12 months after surgery, complaining about instability while walking and lateral pulsion, but is driving his car without difficulties. Other than this, no adverse events were observed.

**Postoperative of peripheral vestibular and auditory function**. The statistical analysis showed that there was no significant difference between pre- and post-operative measures (VOR gain, calorics) or between the two sides postoperatively (VEMPs). The data for each individual patient is shown in Fig. 2. Numerical values and confidence intervals are shown in Table 1. The incidence and direction of SPN did not change between pre- and postoperative testing (Fig. 2j). In 17 of 20 patients (85%), calorics

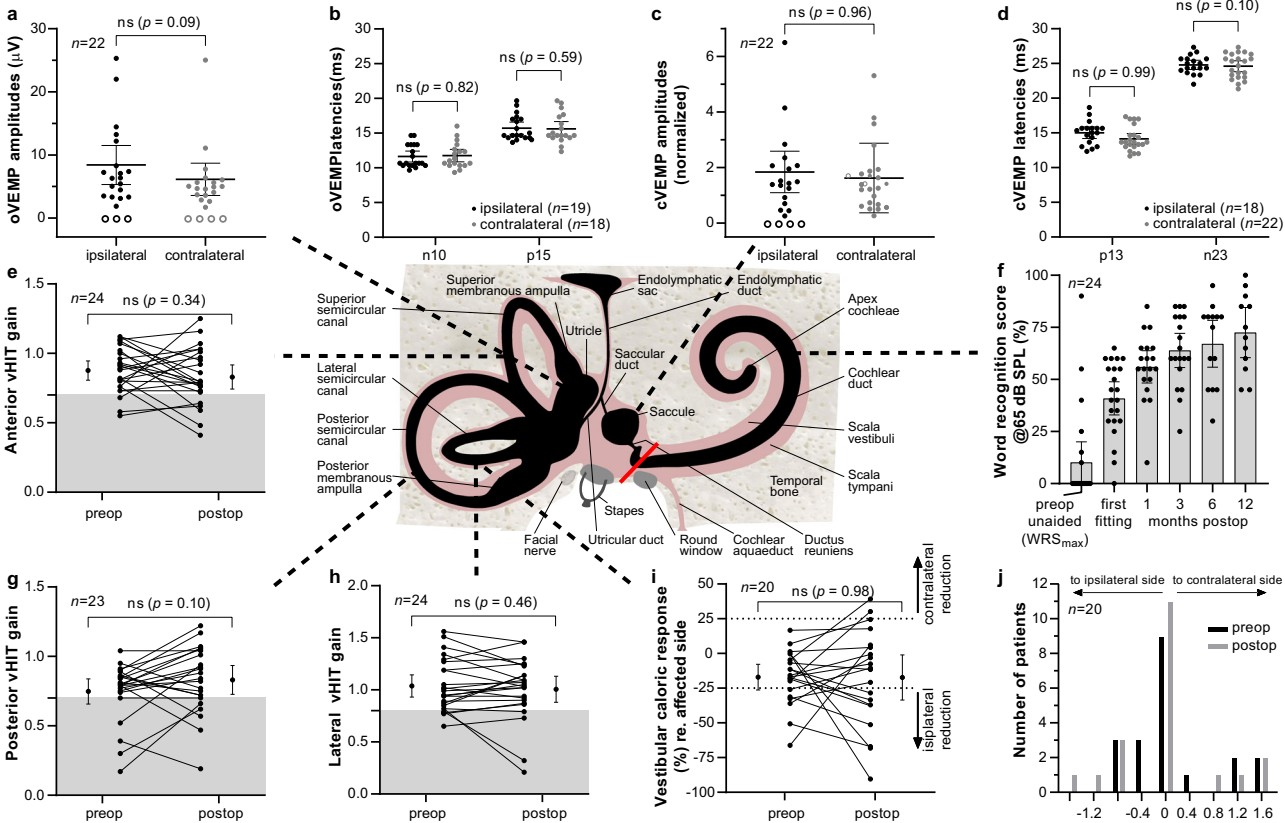

**Fig. 2 Schematic illustration of the inner ear with the five vestibular receptors and the cochlea and results of functional tests before and after surgery.** Postoperative otolith receptor function amplitudes and latencies are shown for utricle (**a**, **b**) and saccule (**c**, **d**) with data points for individual patients, means and 95% confidence intervals. Pre- and postoperative impulse responses of the anterior and posterior semicircular canals (vHIT) are shown on (**e**) and (**g**) with each line showing the data for one patient. The means and 95% confidence intervals are shown adjacent to the data points. The gray areas in the vHIT plots show abnormal results. Fast frequency (vHIT) and low-frequency (caloric) response of the lateral semicircular canal (**h**, **i**) and spontaneous nystagmus (**j**) are shown. 'Ipsilateral' refers to the tumor-affected side and 'contralateral' to the other (healthy) side. Word recognition scores (WRS) for monosyllables at 65 dB SPL with the cochlear implant compared to maximum word recognitions scores before surgery (WRS$_{max}$) are shown on (**f**). The location of the surgical blockage of the vestibular from the auditory system is indicated as a red bar. All statistical comparisons were made with paired two-tailed t-tests. Error bars in (**a**–**i**) show the 95% confidence interval. n number of participants, ns not significant, preop preoperative measurements, postop postoperative measurements, SPL: sound pressure level, VEMP cervical (c) or ocular (o) vestibular-evoked myogenic potentials. vHIT video head impulse test. Schematic illustration adapted from Spalteholz 1920[34].

showed approximately the same or an improved (two patients, 10%) low-frequency response of the lateral semicircular canal compared to the preoperative situation. Only two patients (10%) deteriorated (Fig. 2i). The impulse response of the lateral semi-circular canal, i.e., vHIT gain, was still normal after surgery in 22 of 24 patients (92%). One patient improved slightly but the gain stayed abnormal. Two patients (8%) deteriorated (Fig. 2h). In the anterior and posterior planes, only 3 of 24 (13%) and 2 of 23 (9%) patients showed a deterioration to postoperative pathological gain, respectively. In both planes, two patients improved slightly (8% and 9%, respectively) but the gain stayed abnormal; one patient (4%,) improved to postoperative normal results (Fig. 2e, g). Normal otolith function of the saccule was observed postoperatively in 18 of 22 patients (82%) to bone-conducted vibration (Fig. 2c, d). The utricular function was normal in 19 of 22 patients (86%). Only one of the three patients without oVEMPs had absent responses on the ipsilateral side only, two patients had no responses on both, ipsi- and contralateral side (Fig. 2a, b). Successful hearing rehabilitation with CI and continuous improvement over 1 to 12 months was shown by word recognition at normal speech level compared to maximum word recognition (WRS$_{max}$) before surgery (Fig. 2f).

## Discussion

These results show that vestibular function was largely preserved in most patients after partial or subtotal removal of the cochlea and provide strong objective evidence that the human vestibular labyrinth is far more resistant to trauma than widely believed. The mechanisms and reasons for the observed phenomenon are yet unknown but may be explained by the anatomy of the inner ear fluids spaces and the physiology of mechanoelectrical transduction in the vestibular labyrinth.

The endolymphatic spaces of the cochlea and the vestibular labyrinth are linked by a very thin tube, the ductus reuniens, connecting the cochlear duct and the saccule, with a diameter of <0.2 mm at its narrowest portion (Fig. 2). We consider that this duct is probably sealed by the soft tissue packing after cochlear removal. Due to the blockage of the ductus reuniens, the surgical approach to the cochlea spared major damage to the vestibular labyrinth. Continuous endolymph leakage from the ductus reuniens, larger leakage from the saccule or postoperative infection may explain the loss of vestibular function in 2 of the 27 patients, who on the other hand suffered from vertigo symptoms or attacks already before surgery (the one with persisting instability and lateral pulsion reporting hospitalization for vertigo

**Table. 1 Functional test results before and after partial or subtotal cochlear removal.**

| Measurement | N | Mean | Lower 95% CI of mean | Upper 95% CI of mean | N | Mean | Lower 95% CI of mean | Upper 95% CI of mean |
|---|---|---|---|---|---|---|---|---|
| vHIT gain | | Preoperative | | | | Postoperative | | |
| Lateral semicircular canal | 24 | 1,00 | 0,93 | 1,10 | 24 | 1 | 0,88 | 1,1 |
| Posterior semicircular canal | 23 | 0,75 | 0,66 | 0,84 | 23 | 0,83 | 0,73 | 0,93 |
| Anterior semicircular canal | 24 | 0,88 | 0,81 | 0,95 | 24 | 0,83 | 0,74 | 0,92 |
| Videonystagmography | | | | | | | | |
| Spontaneous nystagmus re. affected side (°/s) | 20 | 0,08 | −0,28 | 0,43 | 20 | −0,03 | −0,39 | 0,34 |
| Caloric irrigation (% response re. affected side) | 20 | −17 | −26 | −7,9 | 20 | −17 | −34 | −1 |
| cVEMP | | | | | | | | |
| p13-n23 Amplitude ipsilateral (EMG normalized) | | | | | 18 | 1.8 | 1.1 | 2.6 |
| p13 Latency ipsilateral (ms) | | | | | 18 | 15 | 14 | 16 |
| n23 Latency ipsilateral (ms) | | | | | 18 | 25 | 24 | 25 |
| p13-n23 Amplitude contralateral (EMG normalized) | | | | | 22 | 1.6 | 1.1 | 2.2 |
| p13 Latency contralateral (ms) | | | | | 22 | 14 | 13 | 15 |
| n23 Latency contralateral (ms) | | | | | 22 | 25 | 24 | 25 |
| oVEMP | | | | | | | | |
| n10-p15 Amplitude ipsilateral (μV) | | | | | 19 | 8.4 | 5.3 | 12 |
| n10 Latency ipsilateral | | | | | 19 | 12 | 11 | 12 |
| p15 Latency ipsilateral | | | | | 19 | 16 | 15 | 17 |
| n10-p15 Amplitude contralateral (μV) | | | | | 18 | 6.1 | 3.6 | 8.7 |
| n10 Latency contralateral | | | | | 18 | 12 | 11 | 13 |
| p15 Latency contralateral | | | | | 18 | 16 | 15 | 17 |
| Speech perception in quiet | | | | | | | | |
| Word Recognition Score (WRS, %), maximum WRS preoperative and at 65 dB SPL postoperative | 24 | 10.2 | 0.4 | 20.0 | 12[a] | 72.5 | 60.5 | 84.5 |

[a]at: 12 months

attacks in his medical history before tumor and CI surgery). However, such a seal deprives the vestibular labyrinth of high potassium ($K^+$) endolymph secreted by stria vascularis and the question becomes how the vestibular labyrinth generates sufficient endolymph to allow continued normal vestibular function.

Mechanotransduction in the vestibular and cochlear sensory cells requires high endolymphatic $K^+$ concentration as an electrochemical motor. The maintenance of endolymph $K^+$ homoeostasis is regulated through a fine balance between secretion and absorption by epithelial cells. The dark cell epithelium and subepithelial melanocytes of the vestibular division of the labyrinth are functionally comparable to the marginal and intermediate cells of the stria vascularis, respectively[18–20]. The molecular repertoire required for endolymph $K^+$ secretion, specifically the transporter and channel proteins NKCC1, KCNQ1, KCNE1, BSND and CLC-K, is shared between the dark cells in the vestibular labyrinth and the marginal cells of the stria vascularis of the cochlea in mammals[21]. Based on the results of our study, we conclude that—since the stria vascularis has effectively been removed with the tumor—there are sufficient endolymph generating cells in the vestibular labyrinth to keep the vestibular sensory organ functioning. This is also supported by findings, that vestibular gravity receptors are highly resistant to acute disruption of endolymph secretion unlike the Organ of Corti in the auditory system[22], were active, highly energy-consuming processes of cochlear outer hair cells guarantee active amplification and non-linear compression of cochlear vibrations (reviewed in ref. [23]). The active processes in the cochlea are likely a major reason for the asymmetry in the resilience of the vestibular labyrinth compared to the cochlea to invasive and traumatic surgery of the respective other system. Although hearing

preservation appears possible after major surgical trauma to the vestibular labyrinth in some cases[6–9,11–13], translabyrinthine skull base procedures are considered non-hearing preservation surgical approaches[24]. Even in lesser traumatic surgeries than translabyrinthine approaches, i.e., in vestibular implantation, hearing is reduced after implantation in a considerable number of patients[25]. An additional explanation for the vestibular labyrinth being more robust to traumatic cochlear surgery than initially thought, might be that vestibular receptors are phylogenetically older than acoustic receptors (hearing)[26].

Preservation of vestibular function after major surgical trauma to the cochlea may depend on the surgical approach. For other surgical approaches for intracochlear schwannoma removal than the one used in our study[27,28]—like "push-through" or "pull through" techniques (also called "pipe cleaner", "beach towel", or "dental floss" techniques[28–31]), no detailed, objective measures on the preservation of vestibular function have been published. We speculate, that similar results on preservation of vestibular function can be obtained with these techniques, although complete tumor removal from the cochlea is less likely and growth of tumor remnants to the vestibule may cause problems later. If the tumor already invades the vestibule with subsequent additional trauma at this location during tumor removal, it appears very unlikely that the function of the vestibular labyrinth can be preserved at all or to such an extent as described as in our study. However, this needs to be investigated in the future.

Although this study shows for the first time in a - considering the rarity of the disease -large number of patients that vestibular function is not only able to be preserved but is most commonly preserved following partial or subtotal cochleoectomy through serial, state-of-the-art vestibular function testing, there are

limitations. Complete functional test results for all five vestibular receptors were not available for all patients. In addition, vestibular function was not evaluated by standardized patient-related outcome measures. This was due to the variability of available tests over the 10-year course of the study and due to the retrospective nature of parts of the study. These aspects need to be addressed in further and follow up-investigations.

In summary, our observations demonstrate that the receptors for the sense of hearing and balance are not only selectively activated, but they are anatomically and physiologically separated enough, so that the receptors for the sense of balance can withstand extensive surgical manipulations to the cochlea. This result has important implications for our understanding of the function and surgery of the inner ear in general and for further advancements in neural interfaces. A current disadvantage of cochlear implantation is the necessity to introduce a multielectrode array through a very small opening in the cochlea which then is pushed forward in the scala tympani around the modiolus with the spiral ganglion cells receiving the electrical stimulation. This results in technical challenges and limitations for electrode array design[32]. Our results open up new possibilities for future designs of neural interfaces for electrical or optical stimulation of the peripheral auditory system.

## Data availability

Source data for the main figures in the manuscript can be accessed as Supplementary Data 1. The remaining data cannot be made publicly available because the ethical approval and the informed consent from the patients included in this study did not cover placing excerpts/copies from patient charts like written records of medical history, or complete electrophysiological curves like full VEMP-, vHIT-, or audiogram traces etc., or third party written reports or images (e.g. MRI, histology etc.) into publicly open repositories. Relevant portions of those data can be accessed from the authors upon relevant ethical approval by contacting the corresponding author on reasonable request.

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

## Acknowledgements

We thank Dr. Gerrit Götze, Dr. Ingmar Seiwerth, Dr. Luise Wagner, and the clinical staff of the department of the corresponding author for patient care. The study was partially supported by project allowances from the German Ministry of Science and Education (BMBF) to SKP (grants: HODOKORT 01KG1427 and ITKORT 01KG2019).

## Author contributions

S.K.P., T.R., and L.F. designed the project and contributed to data acquisition. S.K.P. indicated and conducted the surgeries. S.K.P., T.R., I.S.C., and L.F. analyzed the data. B.H. developed the bone conduction stimulator for recording bone conduction VEMPs. S.K.P., T.R., I.S.C., B.H., and L.F. contributed to interpretating the data and writing the manuscript.

## Funding

## Competing interests

The authors declare no competing interests.
