## [Peer Review File · Communications Medicine]

Reviewers' comments:

Reviewer #1 (Remarks to the Author):

Brief Summary

The study is a large retrospective and prospective case series analyzing vestibular function via vestibular function tests of patient who have undergone partial or subtotal cochleoectomy due to intracochlear schwannoma. The statistical analysis showed that there was no significant difference between pre- and post-operative measures for all tests of the five vestibular organs. The authors postulate that soft-tissue packing of the ductus reuniens helps to preserve endolymph function in the vestibular apparatus. Ability to maintain optimal endolymph may be explained by functional comparability of the dark cell epithelium and subepithelial melanocytes of the vestibular division to the marginal and intermediate cells of the stria vascularis, respectively.

Overall Impression

This is a well-constructed scientific question that supports its conclusion with excellent data. In particular, the authors challenge a common conception of cochlear surgery's negative impact on vestibular function and effectively prove their observations that vestibular function is not only able to be preserved but is most commonly preserved following cochleoectomy through serial, state-of-the-art vestibular function testing. This is a very important and potentially ground-breaking conclusion as it may effectively change cochlear surgery indications and techniques. The authors may consider addition of the evaluation of patient symptoms to support their conclusions even more as ultimately patient experience and self-perception of function is the most important outcome following surgery, though they did define their scope as limited to vestibular function testing. Finally, the authors may consider commenting on different surgical approaches and if that could impact vestibular function otherwise from the singular approach described in this manuscript. Overall, an effective manuscript that opens a new avenue for inquiry into the world of labyrinthine surgery.

Specific Comments

1. Bone-conducted cVEMPs are mentioned lines 210-212: "Normal otolith function of the saccule was observed postoperatively in 18 of 22 patients (82%) to bone conducted vibration (figure 2a,b)." Bone-conducted cVEMPs is a bit of departure from typical protocols and I believe is not mentioned in the Methods.
2. Can the authors comment on the presence of covert/overt saccades on vHITs tracings? Many find that saccades can be a strong predictor of VOR dysfunction even if gain is preserved. Korsager LE, Faber CE, Schmidt JH, Wanscher JH. Refixation Saccades with Normal Gain Values: A Diagnostic Problem in the Video Head Impulse Test: A Case Report. *Front Neurol.* 2017;8:81. Published 2017 Mar 14. doi:10.3389/fneur.2017.00081
3. Please define OSL in Figure 1.
4. Consider labeling the saccular duct in Figure 1.
5. Is it possible to mention symptoms statistics specifically as they are mentioned only qualitatively in the body?
6. Excellent paper and impressive collection of intracochlear cases. Well done!

Reviewer #2 (Remarks to the Author):

COMMSMED-21-0257-T: Plontke et al. Independent vestibular labyrinthine function after major surgical trauma to the human cochlea

This a beautifully clear and well-written report of vestibular function results after cochlear-ablative surgery in 27 cases of cochlear schwannoma. The topic has strong clinical relevance. The methods are well-described and appropriate, as are the conclusions. I have three suggestions for minor revision to strengthen the manuscript – none of which is mandatory but all of which would, I believe, improve the paper:

1. The initial framing of the clinical question/problem of vestibular damage arising from cochlear (surgical) trauma is somewhat overstated. The authors cite several papers reporting vestibular dysfunction after cochlear implant surgery and, in fact about one-third of cochlear implantees report temporary and mild vestibular symptoms. However, there have been hundreds of thousands of cochlear implantations and we have not created a world full of vestibulopaths! Perhaps it would be a more realistic framing of the problem to state that it is well-known that cochlear (implant) surgery can influence vestibular function but, since it is relatively atraumatic, no one has yet probed the resilience of the vestibular labyrinth to more invasive and traumatic surgery. The present study does just this.
2. The Discussion in the manuscript correctly points out the resilience of the vestibular labyrinth to surgical ablation of the cochlea makes some reasonable speculations about the sources of this resilience. On the other hand, there is an emerging literature on vestibular implantation (e.g. by DellaSantina and by Rubinstein) that demonstrates total or near-total loss of hearing in ,the majority of vestibular implant research subjects. It would strengthen the Discussion to compare and contrast these findings.
3. Figure 2, showing a cartoon of the inner ear surrounded by data plots, is drawn with a large cartoon and tiny data presentations. This is backwards. The data are more important than the inner ear schematic. Please shrink the cartoon and enlarge the data plots.

Thank you for the opportunity to review this manuscript.

Steven D. Rauch, MD
Mass Eye & Harvard/Harvard Medical School
Boston, MA, USA

Answer to Reviewers' comments:

We thank the reviewers for their comments, helping to improve the manuscript. Please find our answers below and the changes in the manuscript marked in yellow.

Reviewer #1 (Remarks to the Author):

Brief Summary

The study is a large retrospective and prospective case series analyzing vestibular function via vestibular function tests of patient who have undergone partial or subtotal cochleoectomy due to intracochlear schwannoma. The statistical analysis showed that there was no significant difference between pre- and post-operative measures for all tests of the five vestibular organs. The authors postulate that soft-tissue packing of the ductus reuniens helps to preserve endolymph function in the vestibular apparatus. Ability to maintain optimal endolymph may be explained by functional comparability of the dark cell epithelium and subepithelial melanocytes of the vestibular division to the marginal and intermediate cells of the stria vascularis, respectively.

Overall Impression

This is a well-constructed scientific question that supports its conclusion with excellent data. In particular, the authors challenge a common conception of cochlear surgery's negative impact on vestibular function and effectively prove their observations that vestibular function is not only able to be preserved but is most commonly preserved following cochleoectomy through serial, state-of-the-art vestibular function testing. This is a very important and potentially ground-breaking conclusion as it may effectively change cochlear surgery indications and techniques. The authors may consider addition of the evaluation of patient symptoms to support their conclusions even more as ultimately patient experience and self-perception of function is the most important outcome following surgery, though they did define their scope as limited to vestibular function testing. Finally, the authors may consider commenting on different surgical approaches and if that could impact vestibular function otherwise from the singular approach described in this manuscript Overall, an effective manuscript that opens a new avenue for inquiry into the world of labyrinthine surgery.

→Thank you for this positive comment:

We have now discussed the issue of "...different surgical approaches and if that could impact vestibular function otherwise from the singular approach described in this manuscript..." in a separate paragraph in the discussion section of this article.

While “classical” approaches for cochlear implantation into the normal (i.e., fluid filled cochlea) like cochleostomy and round window approach are aiming on the avoidance of the cochlear duct and are thus, unlikely to exhibit a significant impact on vestibular function:

“...For other surgical approaches for intracochlear schwannoma removal than the one used in our study ^{1,2} – like “push-through” or “pull through” techniques (also called “pipe cleaner”, “beach towel”, or “dental floss” techniques ²⁻⁵), no detailed, objective measures on the preservation of vestibular function have been published. We speculate, that similar results on preservation of vestibular function can be obtained with these techniques, although complete tumor removal from the cochlea is less likely and growth of tumor remnants to the vestibule may cause problems later. If the tumor already invades the vestibule with subsequent additional trauma at this location during tumor removal, it appears very unlikely that the function of the vestibular labyrinth can be preserved at all or to such an extent as described as in our study. However, this needs to be investigated in the future...”

We agree with the reviewer and acknowledge the value of patient experience and self-perception of function (patient reported outcome measures). However, this project was about changes in objective, specific measures of vestibular function. Due to the partially retrospective nature of this study, patient reported outcome measures (e.g. DHI and other) were not systematically documented in a structured way of assessment and a new survey, performed at this time, would result in patient reported outcome measures not matched to the time point of the objective measures. However, we plan to evaluate patient experience and self-perception of function prospectively in further investigations on this topic.

We specified the subjective complaints of the one patient continuously suffering from incompletely compensated ipsilateral loss of vestibular function 12 months after surgery, “...complaining about instability while walking and lateral pulsion, but is driving his car without difficulties.”

Specific Comments

1. Bone-conducted cVEMPs are mentioned lines 210-212: “Normal otolith function of the saccule was observed postoperatively in 18 of 22 patients (82%) to bone conducted vibration (figure 2a,b).” Bone-conducted cVEMPs is a bit of departure from typical protocols and I believe is not mentioned in the Methods.

We understand that bone conducted cVEMPs is different from the typical protocols, so we now explicitly stated that this was done due to intraoperative removal of the incus resulting in significant loss in air conduction.

In addition, air conducted VEMP testing in CI patients bears the risk of false negative results due to (inner ear) conductive loss and response rates to bone conducted stimulation have been reported to be higher compared to air conducted stimulation ⁶ We also specified that both, oVEMPs and cVEMPs, were measured to bone conducted stimulation postoperatively.

We added a very recent reference for the BC transducer used: Fredén Jansson, K.-J., Håkansson, B., Reinfeldt, S., Persson, A.-C. & Eeg-Olofsson, M. Bone Conduction Stimulated VEMP Using the B250 Transducer. *Med. Devices Evid. Res.* Volume 14, 225–237 (2021). ⁷

2. Can the authors comment on the presence of covert/overt saccades on vHITs tracings? Many find that saccades can be a strong predictor of VOR dysfunction even if gain is preserved.

*Korsager LE, Faber CE, Schmidt JH, Wanscher JH. Refixation Saccades with Normal Gain Values: A Diagnostic Problem in the Video Head Impulse Test: A Case Report. *Front Neurol.* 2017;8:81. Published 2017 Mar 14.*

doi:10.3389/fneur.2017.00081

We thank the reviewer for pointing this out. The single case reported by Korsager is of interest. However, although saccades may provide evidence complementing VOR gain and possibly may even identify small changes in VOR gain, at that point, we are rather reluctant to analyze the vHIT VOR data further and in more detail. Substantial evidence of the value of very small corrective saccades is not yet available and there are many variables that can influence saccades, so that they are not a specific indicator of semicircular canal function. In addition, standardized procedures for objective evaluation (quantification and interpretation) are not yet established or uniformly agreed upon.

The main conclusion from our observation is that vestibular function was largely preserved in most patients after subtotal removal of the cochlea. The reason for that conclusion is that the specific, widely accepted test of semicircular canal function, vHIT VOR gain, did not detect a significant difference between average preoperative and postoperative VOR gain across a large number of patients. The graphs show that some patients had increased VOR gain, and some had decreased VOR gain but the important point is that across patients, semicircular canal function was largely preserved, i.e., there was no large, systematic decrease in VOR gain after surgery, which is the main, surprising result of this study.

However, we think that it will be of interest in further studies to investigate whether some individual patients may show slightly different saccadic patterns after surgery.

3. Please define OSL in Figure 1.

 OSL (osseus spiral lamina) was defined in the legend of figure 1.

4. Consider labeling the saccular duct in Figure 1.

→In Fig 1b the saccular duct cannot be seen. In Figure 1a (MRI) we feel that this structure cannot be identified with sufficient certainty. We have labeled the saccular duct in the cartoon in figure 2.

5. Is it possible to mention symptoms statistics specifically as they are mentioned only qualitatively in the body?

→We have added the numbers of patients presenting with hearing loss, vertigo/dizziness/instability, and tinnitus:

“...Preoperatively, 25 patients suffered from complete or functional deafness (indication for CI). Two patients would have benefited from a hearing aid but could not use it due to increase of tinnitus and dysacusia. Hearing loss was due to sudden hearing loss in 6 of 27 (22%) or progressive or recurrent sudden hearing loss in 21 of 27 (78%) patients. Four of 27 patients (15%) reported vertigo attacks in their medical history, with two patients having been hospitalized due to their attacks. Five of 27 patients (19%) reported episodes of instability. Eighteen patients complained about tinnitus (4 patients with pulsating tinnitus), while information about tinnitus was not available for nine patients.”

6. Excellent paper and impressive collection of intracochlear cases. Well done!

→Thank you for this positive comment:

Reviewer #2 (Remarks to the Author):

COMMSMED-21-0257-T: Plontke et al. Independent vestibular labyrinthine function after major surgical trauma to the human cochlea

This a beautifully clear and well-written report of vestibular function results after cochlear-ablative surgery in 27 cases of cochlear schwannoma. The topic has strong clinical relevance. The methods are well-described and appropriate, as are the conclusions. I have three suggestions for minor revision to strengthen the manuscript – none of which is mandatory but all of which would, I believe, improve the paper:

1. The initial framing of the clinical question/problem of vestibular damage arising from cochlear (surgical) trauma is somewhat overstated. The authors cite several papers reporting vestibular dysfunction after cochlear implant surgery and, in fact about one-third of cochlear implantees report temporary and mild vestibular symptoms. However, there have been hundreds of thousands of cochlear implantations and we have not created a world full of vestibulopaths! Perhaps it would be a more realistic framing of the problem to state that it is well-known that cochlear (implant) surgery can influence vestibular function but, since it is relatively

atraumatic, no one has yet probed the resilience of the vestibular labyrinth to more invasive and traumatic surgery. The present study does just this.

→ We agree with the reviewer's comment regarding the initial framing of the clinical problem of vestibular damage following cochlear implantation. We modified the introduction slightly.

2. The Discussion in the manuscript correctly points out the resilience of the vestibular labyrinth to surgical ablation of the cochlea makes some reasonable speculations about the sources of this resilience. On the other hand, there is an emerging literature on vestibular implantation (e.g. by DellaSantina and by Rubinstein) that demonstrates total or near-total loss of hearing in ,the majority of vestibular implant research subjects. It would strengthen the Discussion to compare and contrast these findings.

→ We addressed this now in the discussion section:

“...The active processes in the cochlea are likely a major reason for the asymmetry in resilience of the vestibular labyrinth compared to the cochlea to invasive and traumatic surgery of the respective other system. Although hearing preservation appears possible after major surgical trauma to the vestibular labyrinth in some cases (“old” ref in text body), translabyrinthine skull base procedures are considered non-hearing preservation surgical approaches ⁸. Even in lesser traumatic surgeries than translabyrinthine approaches, i.e., in vestibular implantation, hearing is reduced after implantation in a significant number of patients ⁹. An additional explanation for the vestibular labyrinth being more robust to traumatic cochlear surgery than initially thought, might be that vestibular receptors are phylogenetically older than acoustic receptors (hearing) ¹⁰....”

3. Figure 2, showing a cartoon of the inner ear surrounded by data plots, is drawn with a large cartoon and tiny data presentations. This is backwards. The data are more important than the inner ear schematic. Please shrink the cartoon and enlarge the data plots.

→ Thank you for that tip. We changed the figure accordingly.

Additional literature after answer to reviewer comments:

- 1 Plontke, S. K. An Improved Technique of Subtotal Cochleoectomy for Removal of Intracochlear Schwannoma and Single-stage Cochlear Implantation. *Otol Neurotol* **41**, e891, doi:10.1097/MAO.0000000000002718 (2020).

- 2 Plontke, S. K., Kosling, S. & Rahne, T. Cochlear Implantation After Partial or
Subtotal Cochleoectomy for Intracochlear Schwannoma Removal-A Technical Report.
Otol Neurotol **39**, 365-371, doi:10.1097/MAO.0000000000001696 (2018).
- 3 Aschendorff, A. *et al.* Treatment and auditory rehabilitation of intralabyrinthine
schwannoma by means of cochlear implants : English version. *HNO* **65**, 46-51,
doi:10.1007/s00106-016-0217-8 (2017).
- 4 Ma, A. K. & Patel, N. Endoscope-assisted Partial Cochlectomy for Intracochlear
Schwannoma With Simultaneous Cochlear Implantation: A Case Report. *Otol*
Neurotol **41**, 334-338, doi:10.1097/MAO.0000000000002539 (2020).
- 5 Rahne, T. *et al.* Perioperative Recording of Cochlear Implant Evoked Brain Stem
Responses After Removal of the Intralabyrinthine Portion of a Vestibular
Schwannoma in a Patient with NF2. *Otol Neurotol* **40**, e20-e24,
doi:10.1097/MAO.0000000000002056 (2019).
- 6 Frohlich, L., Wilke, M., Plontke, S. K. & Rahne, T. Bone conducted vibration is an
effective stimulus for otolith testing in cochlear implant patients. *J Vestib Res*,
doi:10.3233/VES-210028 (2021).
- 7 Freden Jansson, K. J., Hakansson, B., Reinfeldt, S., Persson, A. C. & Eeg-Olofsson,
M. Bone Conduction Stimulated VEMP Using the B250 Transducer. *Med Devices*
(*Auckl*) **14**, 225-237, doi:10.2147/MDER.S317072 (2021).
- 8 Chen, B. S. & Brackmann, D. E. in *Comprehensive Management of Vestibular*
Schwannoma (eds M. L. Carlson *et al.*) Ch. 33, 225-234 (Thieme Medical Publishers
Inc., 2019).
- 9 Chow, M. R. *et al.* Posture, Gait, Quality of Life, and Hearing with a Vestibular
Implant. *N Engl J Med* **384**, 521-532, doi:10.1056/NEJMoa2020457 (2021).
- 10 Carey, J. & Amin, N. Evolutionary changes in the cochlea and labyrinth: Solving the
problem of sound transmission to the balance organs of the inner ear. *Anat Rec A*
Discov Mol Cell Evol Biol **288**, 482-489, doi:10.1002/ar.a.20306 (2006).

REVIEWERS' COMMENTS:

Reviewer #1 (Remarks to the Author):

The authors have completed an impressive work. They have addressed all previously highlighted questions and concerns as well as provided more educational resources for this reviewer. Their time, effort and contribution to the field are appreciated. This reviewer cannot identify any more revisions or suggestions and agrees with publication.

Reviewer #2 (Remarks to the Author):

The authors have done an excellent job responding to all queries and comments from the initial review.

-sdr

Steven D. Rauch, MD
Professor and Vice Chair for Clinical Research
Dept. of Otolaryngology - Head & Neck Surgery
Harvard Medical School

Chief, Vestibular Division
Member, Otology/Neurotology Division
Otolaryngology - Head & Neck Surgery Dept.
Mass. Eye & Ear
Mass. General Hospital
Boston, MA 02114

Tel: 617-573-3644

Email: steven_rauch@meei.harvard.edu